# Leveraging Wikidata for Amateur Theatre Research: The WikiFAIR Approach of the P-CITIZENS Amateur Theatre Wiki

Ioanna Papazoglou ⓘ *[1] and Meike Wagner ⓘ †[2]

[1]IT-Gruppe Geisteswissenschaften, LMU Munich

[2]Lehrstuhl für Theaterwissenschaft, LMU Munich

## Abstract

The ERC-funded project P-CITIZENS (Performing Citizenship) explores the social and political roles of amateur theatre in Europe between 1780 and 1850. To support this research, the project has developed the Amateur Theatre Wiki, a platform dedicated to documenting historical and contemporary amateur theatre groups. By adopting a WikiFAIR approach, the project implements an efficient, low-overhead solution that uses Wikidata as the structured-data backend for the Wiki.

The Wiki hosts textual and media content written about and by amateur theatre groups, while Wikidata functions as the repository for structured data—geographic locations, timelines, membership details, affiliations. Leveraging Wikidata's interconnected nature, the project integrates this data into additional knowledge networks, enriching the broader cultural-heritage landscape and enabling extensive data reuse.

An innovative technical feature is the automated rendering of Wikidata information into local MediaWiki pages via Infobox templates. This ensures seamless data presentation for end users while maintaining a centralized external dataset. The separation of content (text and media hosted in the Wiki) from data (stored in Wikidata) enhances reusability, interoperability, and collaborative potential. This case study focuses on strategies for data management (OpenRefine, BeautifulSoup), copyright and community engagement, and impact and accessibility through linked open data.

**Keywords:** Wikidata, Amateur Theatre, WikiFAIR, Digital Humanities, Structured Data

*Ioanna.Papazoglou@itg.uni-muenchen.de
†meike.wagner@lmu.de

# 1   Introduction

As part of the ERC-funded P-CITIZENS (Performing Citizenship) project, we have developed a collaborative platform to document both historical and contemporary amateur theatre. The *Amateur Theatre Wiki* hosts detailed entries on the individuals, organizations, and venues responsible for amateur productions, and it welcomes contributions from volunteers who enrich the site with information about active troupes in their regions.

To jump-start community participation, we employ "red links," a well-established engagement strategy that invites users to create missing articles directly within the Wiki.[1,2] Implementing this feature required assembling an initial roster of amateur-theatre troupes—yet existing public datasets proved sparse, and no specialized registry was available.

In this paper, we first describe our workflow for locating, extracting, and structuring data from diverse web sources, reconciling records against Wikidata, and creating new items for previously undocumented troupes. We then report on our application of the WikiFAIR guidelines to simplify our infrastructure: rather than self-hosting a separate structured database, we integrate Wikidata into our MediaWiki instance, rendering remote statements - such as addresses, foundation dates, and - directly within local infoboxes alongside the full-text articles.

# 2   Extraction of Amateur Theatre Data from Regional Catalogues

Regional theatre federations (or *Dachverbände*) often maintain online catalogues of their member associations (see Figure 1). In Germany, most states provide publicly accessible member lists; however, exceptions such as Baden-Württemberg and Hesse lack suitable formats for automated extraction. For example, Hesse's directory appears only as a dropdown menu on a questionnaire subpage, listing organization names without accompanying metadata. Effective scraping therefore requires at minimum a linked web presence or an associated city name.

To harvest these records, we use Python's BeautifulSoup and the "Instant Data Scraper" Chrome extension. This pipeline locates and parses HTML tables, lists, and form elements, extracting troupe names, locations, and contact details into a structured CSV output suitable for subsequent reconciliation with Wikidata.

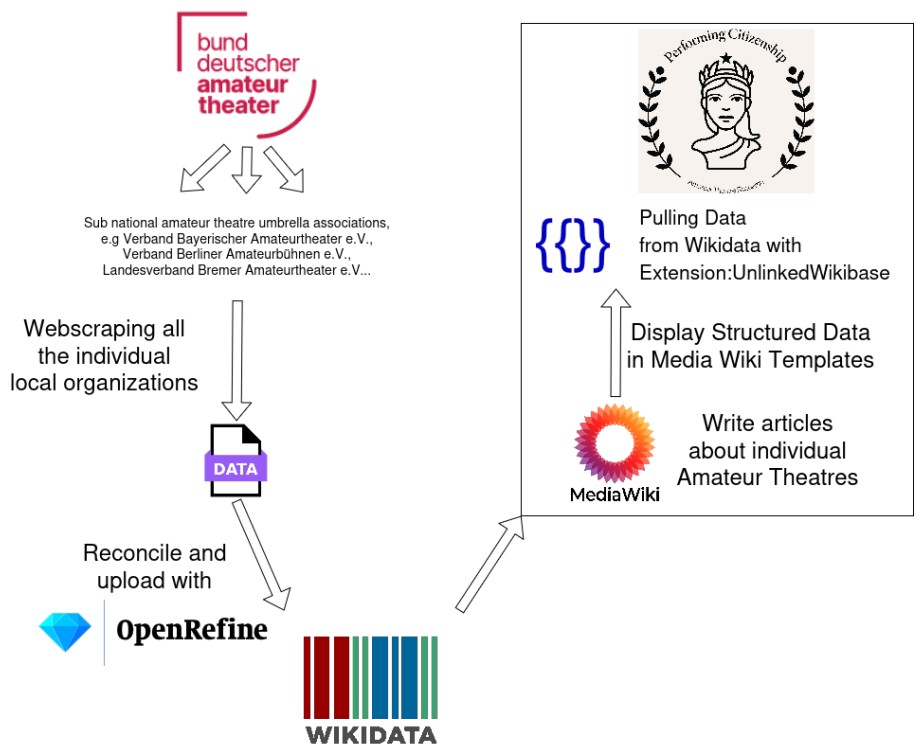

Figure 1: Data extration and ingestion process

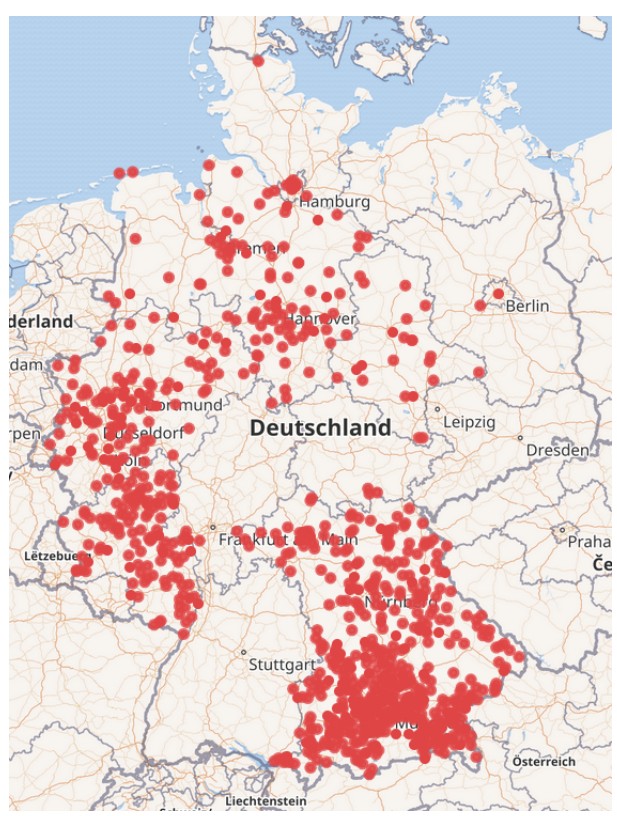

Figure 2: Result of SPARQL query visualizing some Amateur theatre associations in Germany

## 3   Motivation for implementing WikiFAIR recommendations

Having assembled a heterogeneous collection of structured data on amateur theatre associations in Germany, the project faced the challenge of deciding how best to allocate limited resources to present this data within MediaWiki beyond the use of red links. WikiFAIR offers guidelines, developed in collaboration with the WikiKULT Community of Practice, for implementing FAIR principles in research projects with constrained budgets and manpower. These guidelines recommend outsourcing knowledge infrastructure to existing Wikimedia projects in order to reduce the complexity of a project's technical stack. Although we deployed a simple self-hosted MediaWiki instance via Docker and managed with Ansible, we realised early on that running a separate Wikibase deployment to manage the newly acquired structured data would require substantial additional effort. Consequently, we followed the WikiFAIR recommendation to reintegrate our dataset into Wikidata proper. This approach conserved considerable project resources that would otherwise have been spent developing a custom ontology and maintaining multiple components of the Wikibase ecosystem. Moreover, by leveraging Wikidata's API, long-term hosting, and backup infrastructure, our data automatically inherits FAIR compliance through an established, sustainable platform.

## 4   Upload process

To upload data to Wikidata, we first check for existing amateur theatre troupes by running a reconciliation request[3] through OpenRefine. This step prevents the creation of duplicate items for well-known associations that already appear in Wikidata due to existing Wikipedia articles. During reconciliation, we apply an import schema based on the properties listed in Table 1. For long-term citability, we link to Archive.org snapshots of the original member lists whenever possible (see Figure 3). Finally, we visualize the newly uploaded data on a map, displaying the coordinates of each troupe's nearest city (Figure 2).

## 5   Automated Infobox Rendering via Unlinked Wikibase

Because we elected not to deploy our own Wikibase instance, the standard Wikibase Client and Repository extensions—which require direct database access—cannot render structured data in MediaWiki pages. Consequently, traditional Wikidata infobox templates, which depend on

Figure 3: References on some statements uploaded to Wikidata

| Property | Description | Example Value |
|----------|-------------|---------------|
| P31 | instance of | Q2416217 (theatre troupe) |
| P571 | inception date | date of founding |
| P1448 | official name | full legal name of the association |
| P101 | field of work | Q455695 (amateur theatre) |
| P17 | country | Q183 (Germany) |
| P463 | member of | e.g. Q2513550 (Verband Bayerischer Amateurtheater) |
| P159 | headquarters location | nearest city (from contact address) |
| P1454 | legal form | Q9299236 (registered association) |
| P856 | official website | URL of the troupe or association |

Table 1: Added properties to Wikidata

parser functions tied to a locally linked Wikibase, are inoperable on our platform.

Several alternatives exist:

- Embedding live SPARQL query results from the Wikidata Query Service via an `<iframe>`, which can display maps or tables but cannot integrate with template logic.

- Adopting a federation extension such as LinkedWiki, which introduces its own parser functions and configuration overhead.

- Using a Lua-based solution that fetches and caches remote entities, provided by the UnlinkedWikibase extension.

We selected UnlinkedWikibase both for its minimal footprint and for its compatibility with exist-

ing infobox templates. Once installed, the extension adds parser functions—such as `#entity`, `#props`, and `#statements`—that retrieve labels, claims, and sitelinks from Wikidata over HTTP and cache them locally for a configurable interval. No database schema changes are required, and our Lua modules need only specify a Q-ID to populate an infobox dynamically.

This approach preserves the familiar MediaWiki templating workflow, reduces maintenance overhead, and ensures that all structured data remain up-to-date, fully FAIR-compliant, and seamlessly integrated alongside our full-text articles.

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
