# OpenReview forum: "Wikidata as a Backend for Research MediaWikis: A Case Study from the P-CITIZENS Project in documenting Amateur Theatre"
_wikimedia.it/Wikidata_and_Research/2025/Conference — WD&R Paper_

### Official Review · ~Silvia_Bruni1 · 2025-01-10
**Wikidata for Documenting Historical Amateur Theatre**

**Originality:** 5
**Impact:** 4
**Confidence:** 4

**Review:**

The project is interesting because it focuses on the history of a broad constellation of institutions and individuals connected to amateur theatre, which has been documented in a fragmented and not easily accessible manner. The connection with a European project (ERC-funded project P-CITIZENS - Performing Citizenship explores the social and political roles of amateur theatre in Europe between 1780 and 1850) demonstrates the versatility of Wikidata and its potential uses. The infobox, designed to make the information more easily readable, will enhance the usability of the data. The project lends itself to open science activities and could serve as a replicable model.
The Amateur Theatre Wiki platform is still empty at the time of review. Examples can be viewed.

**Compliance:**

4

**Scientific Quality:**

4

---

### Official Review · ~Rossana_Morriello1 · 2025-01-11
**An extremely important avenue for Wikidata and research**

**Originality:** 5
**Impact:** 5
**Confidence:** 3

**Review:**

The project is very interesting and useful for studies on theatre and performing arts. Its relevance is confirmed by being an ERC project. The use of Wikidata to support national and international research projects, as in this case, is an extremely important avenue for Wikidata, in line with the theme of collaboration with the academic research community proposed by the conference.

**Compliance:**

5

**Scientific Quality:**

4

---

### Decision · Program_Chairs · 2025-02-05

Accept (Paper)